# Meningitis without Rash after Reactivation of Varicella Vaccine Strain in a 12-Year-Old Immunocompetent Boy

**DOI:** 10.3390/vaccines11020309

**Published:** 2023-01-30

**Authors:** Sibylle Bierbaum, Veronika Fischer, Lutz Briedigkeit, Claudius Werner, Hartmut Hengel, Daniela Huzly

**Affiliations:** 1Center for Microbiology and Hygiene, Institute for Virology, Medical Center, University of Freiburg, 79104 Freiburg, Germany; 2Faculty of Medicine, University of Freiburg, 79110 Freiburg, Germany; 3German Consulting Laboratory for HSV and VZV, Medical Center, University of Freiburg, 79104 Freiburg, Germany; 4Helios Kliniken Schwerin, 19055 Schwerin, Germany

**Keywords:** varicella zoster virus, varicella vaccine, zoster sine herpete

## Abstract

Acute neurologic complications from Varicella-Zoster-Virus reactivation occur in both immunocompromised and immunocompetent patients. In this report, we describe a case of a previously healthy immunocompetent boy who had received two doses of varicella vaccine at 1 and 4 years. At the age of 12 he developed acute aseptic meningitis caused by vaccine-type varicella-zoster-virus without concomitant skin eruptions. VZV-vaccine strain DNA was detected in the cerebrospinal fluid. The patient made a full recovery after receiving intravenous acyclovir therapy. This disease course documents another case of a VZV vaccine-associated meningitis without development of a rash, i.e., a form of VZV infection manifesting as “zoster sine herpete”.

## 1. Introduction

Varicella-Zoster-Virus (VZV) is a member of the herpesvirus family causing varicella during primary infection. Live attenuated varicella vaccines, derived from the Oka strain of VZV, have clinical efficacy for the prevention of varicella [1]. The USA introduced universal varicella vaccination in the national immunization program in 1995, Australia in 2000 and Germany in 2004. Consistently, the countries have subsequently experienced a substantial decrease in disease burden by varicella [2,3,4,5]. In the European Union recommendations for varicella vaccination vary. In some countries (Germany, Greece and few regions of Italy and Spain) varicella vaccination is part of the routine vaccination program during childhood, but the majority of the countries only recommend vaccination of susceptible adolescents, health care workers or high-risk groups [6]. The shifting epidemiology of varicella due to the varicella vaccination programs has made it more challenging for healthcare workers to recognize the disease, because vaccinated persons often show mild and atypical symptoms. In addition, VZV immunity testing will become increasingly challenging since most standard serological tests are not sensitive enough to detect low antibody titers, but vaccination typically results in lower levels of IgG antibody than wild-type (wt) infection [7].

Reactivation of latent VZV genomes from neuronal cell in sensory ganglia, e.g., of the dorsal root and trigeminal nerve results in viral spread along axons and herpes zoster (shingles) which is typically characterized by a vesicular rash limited to a specific dermatome and unilateral radicular pain. Recent studies indicated that vaccinated children have a significantly lower incidence of herpes zoster than unvaccinated children [8,9,10]. Acute neurologic complications from VZV reactivation are frequent and include pain, diffuse or local encephalitis, aseptic meningitis, myelitis and various pattern of cerebrovascular disease [11]. Even if most cases of shingles and their complications are the result of reactivation with wild type VZV, serious adverse events with VZV vaccine strain have been documented including in immunocompetent children [12,13,14,15,16,17]. The differentiation of the live attenuated Oka varicella vaccine from wild-type VZV has become eminently clinically important in certain situations. Testing can be used to assess varicella vaccine effectiveness by identifying cases of breakthrough varicella caused by wt virus [18,19]. Strain discrimination testing is also used to document adverse events associated with the VZV vaccine strain, in particular in case of suspected shingles [19]. However, data on vaccine safety from the USA show that serious adverse events after varicella vaccination are rare, reinforcing the favorable safety profile of the vaccines [20]. In this report, we present a case of a previously immunized immunocompetent child who developed VZV vaccine-associated meningitis. Cases of VZV vaccine-associated meningitis have been reported in previous studies. Typically, these cases were accompanied by a rash [21] or information on this clinical sign was missing from the report [22]. Although several similar cases in children have already been described, this is only the second example without development of a rash. Unlike our patient, however, the first published case from Japan was caused by the Oka/Biken strain derived varicella vaccine [23].

## 2. Case Description

A healthy 12-year-old boy presented to the emergency department with history of severe headache for 2 days, photophobia and vomiting, but no fever. The night before hospitalization symptoms intensified and neck stiffness occurred. There was no history of recent contact with sick people or exposures to varicella. The patient’s immunization status was complete. According to national vaccination recommendations the child had received two documented doses of varicella vaccine at the age of one (Varivax^®^) and four (Priorix-Tetra^®^) years. Cardiac, pulmonary and abdominal examination findings were normal. Magnetic resonance imaging findings were normal. At admission, laboratory analysis of cerebrospinal fluid showed a slight increase of total protein-level (628 mg/L) and lactic acid-level (2.27 mmol/L) as well as a mild to moderate pleocytosis (CSF leukocyte count 2.91 × 10^8^ cells/L). Blood counts revealed a transient lymphocytopenia and a neutrophilia. Results of serum chemistry, including measurements of liver enzyme levels (aspartate aminotransferase level, 32.4 U/L; alanine aminotransferase level, 18.6 U/L), were normal. Enzyme immunoassays were positive for VZV-specific IgM- and IgG-antibodies, indicating active VZV infection including VZV reactivation [24].

PCR analysis of the cerebrospinal fluid was positive for varicella-zoster virus DNA (14,000 copies/mL) and negative for enterovirus and herpes simplex virus by real-time polymerase chain reaction; culture results were negative for bacteria.

Differentiating wild-type VZV strains from the Oka vaccine strain can be achieved only by molecular genotyping methods. The development of PCR based methods for VZV strain differentiation has been hampered by the fact that the VZV genome is highly conserved. However, a number of single nucleotide polymorphisms (SNPs) have been described that differentiate Oka-derived VZV vaccine strains from wild-type VZV strains [25,26,27,28]. Different methodologies have been described which rely on those polymorphisms such as melting curve analysis [29], long-distance PCR [30], single-strand conformation polymorphism analysis [31] and fluorogenic PCR (TaqMan assay) [32]. Also PCR amplification of selected VZV DNA sequences, followed by restriction enzyme digestion for detection of sequence variations or sequence analysis, is a sensitive and reliable approach [19,27,28,31,33,34]. Only recently, the differentiation between vaccine and wild-type strains of VZV was made by metagenomic next generation sequencing [22]. In our case PCRs with primers for single nucleotide polymorphisms (SNPs) located in open reading frame (ORF) 6 (SNP 5745), ORF 54 (SNP 94167) and ORF 62 (SNPs 105544, 105705, 106262, 107136, 107252, 108111) of the varicella-zoster virus genome were used to discriminate vaccine from wild-type strains [19,35]. PCR products were sequenced and evaluated at the indicated variable positions. Only three of the analysed variants carried wildtype alleles (SNP 5745 (data shown in Figure 1A), SNP 94167 (data not shown) and SNP 107136 (data shown in Figure 1D). Five of the analysed variants (SNPs 105544, 105705, 107252 (data shown in Figure 1B,C,E) and 106262, 108111 (data not shown)) carried bona fide vaccine markers, including two particularly important fixed alleles in ORF 62 (105705 and 107252) [36].

Altogether, the SNP profile fully matched with the nucleotide sequence of Oka/Merck varicella vaccine strain as published by Tillieux et al. [37]. On this basis, a regular notification of a suspected case of vaccination complication was initiated informing the Paul-Ehrlich-Institute, the national institution responsible for the pharmacovigilance of vaccines.

Extensive and detailed immune diagnostics revealed normal numbers, percentages and ratios of leukocytes, lymphocytes, monocytes, granulocytes and peripheral T and B cell subsets, including CD4- and CD8- T cells. The analysis was carried out in the Institute of Laboratory and Transfusion medicine at the Helios Hospital Schwerin (data shown in Table 1).

The patient received intravenous acyclovir for 10 days during hospitalization. Meanwhile symptoms improved significantly. The child made an uneventful recovery and was discharged from the hospital without medications.

## 3. Discussion

Zoster typically presents as a painful skin rash leading to blisters within a dermatomal distribution. The availability of sensitive PCR-based detection methods has greatly extended the clinical spectrum of acute and chronic diseases associated with VZV [38]. A growing number of case reports as well as large case series of adults but also children have documented various neurologic complications without skin manifestations in non-immunocompromised persons caused by reactivated VZV that was found in cerebrospinal fluid [39,40]. Diseases induced by reactivated VZV replication occurring with restriction to the CNS include encephalitis, aseptic meningitis, cranial nerve affection and cerebrovascular disease with stroke, conditions summarized as “zoster sine herpete” due to the absence of VZV dissemination to the skin. In such cases, the resulting clinical picture is lacking the rash, which constitutes the classical full-blown zoster disease. Implementation of universal varicella vaccination programs over the last years has introduced attenuated VZV viruses into the population. Their biological phenotypes deviate from wt VZV, particularly regarding their ability to establish replication in the skin [41]. As wt VZV, Oka derived attenuated vaccine strains establish latent infection in neurons and seem to exhibit a moderately reduced ability to cause herpes zoster later in life [42]. However, numerous studies have shown that vaccinated children are 4–12 times less likely to get herpes zoster than unvaccinated persons and the disease often shows a milder clinical course in vaccinated people [8,43]. Vaccination against the varicella-zoster virus is recommended, even if side effects of vaccination occasionally occur, since infection can lead to serious complications and death [44,45]. As studies have shown, the safety profile of varicella vaccine is excellent [12,46].

Based on the presence of single nucleotide polymorphisms (SNP) profiles in various VZV open reading frames (ORF) including ORF 62 coding for the transactivating major immediate early protein diagnostic differentiation of vaccine-type VZV from wt VZV can be achieved [37,47,48]. Moreover, Oka-derived licensed vaccines Varivax^®^ (Merck) and Varilix^®^/Priorix-Tetra^®^ (GSK) significantly differ in their vaccine/wt allele frequency [47,49]. The GSK vaccine exhibits a substantial loss of wt alleles, particularly in ORF 62, and a documented lower diversity of genomic variants [48,49]. The genetic differences are likely to account for distinctions with regard to rash-associated VZV genomes as a consequence of in vivo selection but also vaccine effectiveness against breakthrough varicella [50]. Both findings support the notion that the Merck vaccine possesses more wt-like features. In the light of these reports it is plausible that the reactivated VZV genomes found in the CSF of our patient could be attributed to the first vaccine dose, i.e., Varivax^®^ rather than Priorix-Tetra^®^ which was used as a second dose. A limitation of our study is the limited number of SNPs analyzed and their location within the VZV genome. Recombination between wt VZV strains is widely documented, raising the theoretical possibility that the reactivating VZV genome in our patient might have been originated from a recombination event of the Merck vaccine strain and a superinfecting wt VZV strain. The very existence and clinical relevance of VZV vaccine strain recombination events is a matter of recent debate [24,51] Recently, whole genome sequencing was applied to exclude the presence of wt VZV sequences being involved in a case of acute retinal necrosis after vaccination with the ZostaVax/Merck vaccine strain [52]. Such comprehensive approaches should be employed in suspected cases of VZV meningitis in vaccinees in the future.

## 4. Conclusions

The reported case challenges the common notion that skin manifestations represent a reliable clinical indicator for VZV disease and suggests considering VZV a possible cause of aseptic meningitis in suspected cases with varicella vaccination history. One may speculate that attenuated VZV strains could tend to produce incomplete clinical pictures without rash due to their impairment in replication and spread. Appropriate testing to identify VZV in CSF and sequence analysis in future cases will be important to establish the incidence of this probably rare vaccine complication.

## Figures and Tables

**Figure 1 vaccines-11-00309-f001:**
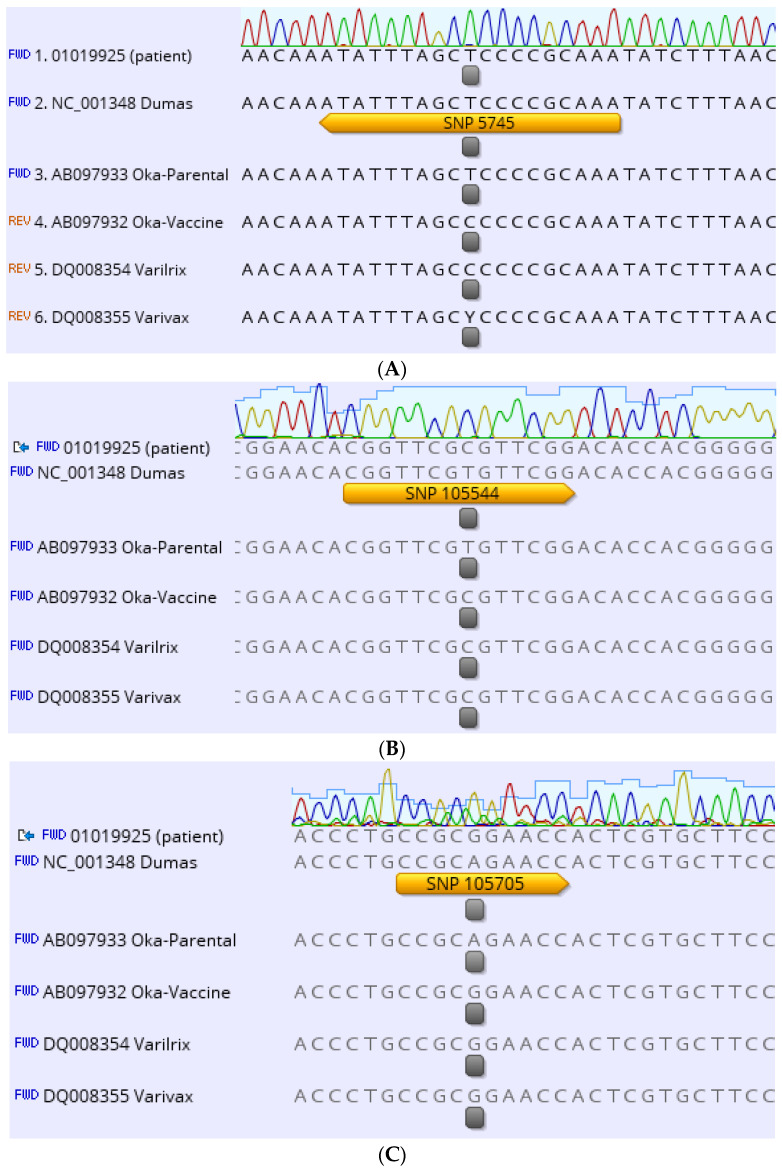
(**A**–**E**): Analysis of SNPs (highlighted by yellow marked arrow) using data from VZV ORF 6 (SNP 5745) and 62 (SNPs 105544, 105705, 107136, 107252). Chromatograms are shown on top. The sequence positions were compared with the sequence of the European VZV strain Dumas as a reference (Gen Bank Accession no. NC_001348). The SNPs are located at the positions marked in gray.

**Table 1 vaccines-11-00309-t001:** Results of patient’s haematological analysis.

Parameter	Patient	Reference Range
White Cell Count	5700 cells/µl	4500–13,500 cells/µL
Lymphocytes	32.3%	20.4–51.1%
Monocytes	9.1%	1–14%
Neutrophil granulocytes	50.1%	42.2–75.2%
B-lymphocytes (CD19+)	18.6%	8–24%
T-lymphocytes (CD3+)	73.1%	52–78%
T-helper cells (CD3+CD4+)	43.6%	25–48%
Cytotoxic T cells (CD3+CD8+)	24%	9–35%
CD4/CD8 ratio	1.8	0.9–3.4

## Data Availability

All of the relevant data are included in the paper.

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
