# Peer review of "Meningitis without Rash after Reactivation of Varicella Vaccine Strain in a 12-Year-Old Immunocompetent Boy"

_vaccines, 2023, doi:10.3390/vaccines11020309_

Round 1
Reviewer 1 Report
In the submitted manuscript, authors present a single case study of a patient presenting with neurological symptoms. Moving through the case description, authors hone in on Varicella Zoster virus as the potential cause of symptoms. They then present evidence suggesting the source of VZV was the initial vaccine dose the patient received many years prior.
Overall, this is a straightforward case-study of an under-studied yet not entirely infrequent outcome. It is a critical evaluation of both the understanding and analysis of VZV infection complications classified as “zoster sine herpete”. The biggest issue is the lack of rigorous testing of viral genomic sequence and the incompletely supported claim that the initial vaccine is the cause of illness. I strongly recommend that authors refer to an excellent and cumulative resource of interpretation of this research topic:
“Impact of Varicella Vaccine on Varicella-Zoster Virus Dynamics” D. Scott Schmid and Aisha O. Jumaan
This text provides some context for the serological results and on the standard for genotyping. It will provide important understanding whether presented illness is caused by vaccine induced reactivation or from a recombination between the vaccine strain and a wild-type circulating VZV strain. Specific comments and suggestions are below:
1) Line 70 – “Enzyme immunoassays…” It is a missed opportunity to interpret the serology in light of the standard “low IgG” detection from vaccinated patients referred in the introduction. Many naïve readers assume IgM might be associated with primary infection. As discussed in the Schmid reference above, IgM is a common hallmark during most VZV reactivation events, independent of vaccine status. So high IgM and IgG is consistent with an active VZV infection stimulating a strong immune response. A point in the case study or discussion is justly warranted.
2) Line 73 – 81 “PCR primers for single nucleotide polymorphisms…” This is a very important point and needs to be expanded upon. Just because 2 SNPs match the vaccine strain doesn't mean the whole virus is vaccine derived. As it reads, many of the analyzed SNPs do not indicate vaccine strain. It is quite possible that you are detecting recombinant genomes between wt and vaccine derived variants. Or that some kind of recombination between vaccine strains resulted in a virus with more pathogenic potential. As Schmidt et al point out:
“It can no longer be considered adequate to confirm vaccine adverse events using the targeted analysis of nucleotide variability at several single nucleotide polymorphisms (SNP). Extensive SNP analysis combined with routine genotyping will now be necessary to be certain that disease is attributable to Oka vaccine.” (pg 213, col 1).
At minimum, the sequence data from the PCR amplified regions and/or a clear visual presentation of the resulting homology for all SNP’s tested needs to be presented. The major novelty of this case study is the attribution of illness due to reactivation of the attenuated vaccine. Therefore this evidence needs to be the strongest to support such a conclusion.
3) Line 106 “their pathogenic potential..” The breadth of recognized pathogenic potential, as noted in the earlier parts of this paragraph, is increasing for both wild-type and vaccine strains. To suggest that risk of severe illness was not evaluated is not appropriate. The Merck vaccine has been in use since the 1980’s and there are numerous studies evaluating risk of childhood meningitis that can counter such an implication of a lack of study of this topic.
4) Line 128 “ that “attenuated” VZV strains…” Authors are implying that the vaccine strains are not sufficiently attenuated or safe. I think it behooves the authors as medical professionals to weigh such statements in light of the pronounced health benefits of the VZV vaccine. While all attenuated viruses have an inherent risk, that risk is more than offset by the benefits of vaccination to the population. Removal of this insinuation is strongly recommended and inclusion of the benefits of VZV vaccination at the conclusion is even more strenuously requested.
Author Response
Manuscript: vaccines-2111458, Bierbaum et al.
Point by point reply by the authors
We are grateful to both of the reviewers for their very careful evaluation of our case study and their helpful suggestions and comments.
We respond to the issues raised by the reviewers in a point-by-point fashion. For orientation, we have responded to the reviewers’ comments in italic.
Reviewer #1
In the submitted manuscript, authors present a single case study of a patient presenting with neurological symptoms. Moving through the case description, authors hone in on Varicella Zoster virus as the potential cause of symptoms. They then present evidence suggesting the source of VZV was the initial vaccine dose the patient received many years prior.
Overall, this is a straightforward case-study of an under-studied yet not entirely infrequent outcome. It is a critical evaluation of both the understanding and analysis of VZV infection complications classified as “zoster sine herpete”. The biggest issue is the lack of rigorous testing of viral genomic sequence and the incompletely supported claim that the initial vaccine is the cause of illness. I strongly recommend that authors refer to an excellent and cumulative resource of interpretation of this research topic:
“Impact of Varicella Vaccine on Varicella-Zoster Virus Dynamics” D. Scott Schmid and Aisha O. Jumaan
This text provides some context for the serological results and on the standard for genotyping. It will provide important understanding whether presented illness is caused by vaccine induced reactivation or from a recombination between the vaccine strain and a wild-type circulating VZV strain. Specific comments and suggestions are below:
Reply: We are grateful for the reviewer’s comments and hints which supported us to improve the case study.
1) Line 70 – “Enzyme immunoassays…” It is a missed opportunity to interpret the serology in light of the standard “low IgG” detection from vaccinated patients referred in the introduction. Many naïve readers assume IgM might be associated with primary infection. As discussed in the Schmid reference above, IgM is a common hallmark during most VZV reactivation events, independent of vaccine status. So high IgM and IgG is consistent with an active VZV infection stimulating a strong immune response. A point in the case study or discussion is justly warranted.
Reply: We agree with the reviewer that the detection of VZV-IgM in our patient should be further commented. With reference to the review by Schmid and Jumaan we state now (line 72-73) “Enzyme immunoassays were positive for VZV-specific IgM- and IgG-antibodies, indicating active VZV infection including VZV reactivation [22].
2) Line 73 – 81 “PCR primers for single nucleotide polymorphisms…” This is a very important point and needs to be expanded upon. Just because 2 SNPs match the vaccine strain doesn't mean the whole virus is vaccine derived. As it reads, many of the analyzed SNPs do not indicate vaccine strain. It is quite possible that you are detecting recombinant genomes between wt and vaccine derived variants. Or that some kind of recombination between vaccine strains resulted in a virus with more pathogenic potential. As Schmidt et al point out:
“It can no longer be considered adequate to confirm vaccine adverse events using the targeted analysis of nucleotide variability at several single nucleotide polymorphisms (SNP). Extensive SNP analysis combined with routine genotyping will now be necessary to be certain that disease is attributable to Oka vaccine.” (pg 213, col 1).
At minimum, the sequence data from the PCR amplified regions and/or a clear visual presentation of the resulting homology for all SNP’s tested needs to be presented. The major novelty of this case study is the attribution of illness due to reactivation of the attenuated vaccine. Therefore this evidence needs to be the strongest to support such a conclusion.
Reply: We agree with the reviewer that recombination events between VZV wt and vaccine strains are not excluded by our SNP analysis. We mention and discuss this limitation of our study in the discussion part (see line 157-167). Moreover, we have included figures highlighting the sequence analysis of three key SNPs (see new figure A, B and C). The comment of the reviewer has prompted us thoroughly screening the available literature. Unfortunately, we were not able to identify a peer-reviewed report of such a case demonstrating recombination between VZV Oka and VZV wt as a cause of reactivation and shingles. However, we refer to available original and review papers on the subject matter (new references [22,25,40]). In future, next generation sequencing approaches will be suitable for detecting such recombination events of VZV vaccine genomes. For this reason, we mention a case report in which this method was successfully used to identify the authentic Merck vaccine virus as the cause of ARN (new reference [41]).
3) Line 106 “their pathogenic potential..” The breadth of recognized pathogenic potential, as noted in the earlier parts of this paragraph, is increasing for both wild-type and vaccine strains. To suggest that risk of severe illness was not evaluated is not appropriate. The Merck vaccine has been in use since the 1980’s and there are numerous studies evaluating risk of childhood meningitis that can counter such an implication of a lack of study of this topic.
Reply: We agree with the reviewer and have added the following statement: (see line 137 and further): However, numerous studies have shown that vaccinated children are 4-12 times less likely to get herpes zoster than unvaccinated persons and the disease often shows a milder clinical course in vaccinated people [8,32]. Vaccination against the varicella-zoster virus is recommended, even if side effects of vaccination occasionally occur, since infection can lead to serious complications and death [33,34]. As studies have shown, the safety profile of varicella vaccine is excellent [12,35].
4) Line 128 “ that “attenuated” VZV strains…” Authors are implying that the vaccine strains are not sufficiently attenuated or safe. I think it behooves the authors as medical professionals to weigh such statements in light of the pronounced health benefits of the VZV vaccine. While all attenuated viruses have an inherent risk, that risk is more than offset by the benefits of vaccination to the population. Removal of this insinuation is strongly recommended and inclusion of the benefits of VZV vaccination at the conclusion is even more strenuously requested.
Reply: We agree and have deleted the misleading quote sign (see line 173).
Reviewer 2 Report
The authors reported a case of aseptic meningitis and zoster sin herpete caused by VZV vaccine strain. This case cannot be distinguished between wild and vaccine strains of VZV without detailed analysis, but SNPs hav e been used to determine this. This case report is an important report for future consideration of the safety of the Varicella vaccine. However, there are already similar existing reports and this case report is not novel. It might be acceptable if you include a literature review of CNS complications due to VZV vaccine based on this case report and future remedies for this adverse reaction.Author Response
Reviewer #2
The authors reported a case of aseptic meningitis and zoster sin herpete caused by VZV vaccine strain. This case cannot be distinguished between wild and vaccine strains of VZV without detailed analysis, but SNPs hav e been used to determine this. This case report is an important report for future consideration of the safety of the Varicella vaccine. However, there are already similar existing reports and this case report is not novel. It might be acceptable if you include a literature review of CNS complications due to VZV vaccine based on this case report and future remedies for this adverse reaction.
Reply: We thank the reviewer for his/her comment. We have searched in the literature for additional reports and discuss the cases (see line 52-54) we found, thus providing the reader with a kind of scoping review of cases with VZV vaccine associated meningitis. See new references [19,20].
Round 2
Reviewer 1 Report
The authors are commended for the revisions and data inclusion in response to the peer review comments. The changes to the text do allay many of my concerns.
The only suggestion I have pertains to the presentation of the SNP sequencing data. As it stands, the presentation is confusing, does not adequately convey the information, and lacks in quality. The sequence viewer that these images are screen shot from has too many “options” turned on. We do not need to see the open reading frames (yellow and green bars) or the consensus sequence at top. Secondly, the order of the different aligned genomes needs to be consistent. It should be Dumas, Oka-Parental, then the three vaccine strains. There are a number of better alignment tools that can be employed, examples of which can be found for free at EMBL (https://www.ebi.ac.uk/services)
Secondly, the chromatograms are an important element that needs to be retained. Figure B needs improved and consistent display of the chromatograms. I have concerns about the background in these presented sequences, especially in SNP 105544. There is a minor “T” band that is masked in the base-calling software. While this may be due to contaminating sequence or a bad read, it could suggest a mixed population of nucleic acids at this region. Repeating the sequencing or selecting clearer reads would either confirm or allay these concerns.
Importantly, the data displayed needs to be consistent in color tone and aspect ratio, both of which are mismatched in Figure C. Any colored boxes need to be explained with a legend and be consistently used in all panels.
Reviewer 2 Report
Revised manuscripts have been appropriately revised in response to reviewers' comments.
